# Desert Endemic Plants in Algeria: A Review on Traditional Uses, Phytochemistry, Polyphenolic Compounds and Pharmacological Activities

**DOI:** 10.3390/molecules28041834

**Published:** 2023-02-15

**Authors:** Hadia Hemmami, Bachir Ben Seghir, Soumeia Zeghoud, Ilham Ben Amor, Imane Kouadri, Abdelkrim Rebiai, Abdelmalek Zaater, Mohammed Messaoudi, Naima Benchikha, Barbara Sawicka, Maria Atanassova

**Affiliations:** 1Department of Process Engineering and Petrochemical, Faculty of Technology, University of El Oued, El Oued 39000, Algeria; 2Renewable Energy Development Unit in Arid Zones (UDERZA), University of El Oued, El Oued 39000, Algeria; 3Laboratory of Industrial Analysis and Materials Engineering (LAGIM), University May 8, 1945, Guelma 24000, Algeria; 4Department of Process Engineering, Faculty of Technology, University May 8, 1945, Guelma 24000, Algeria; 5Department of Chemistry, Faculty of Exact Sciences, University of El Oued, El Oued 39000, Algeria; 6Biodiversity Laboratory and Application of Biotechnology in Agriculture, University of El Oued, El Oued 39000, Algeria; 7Department of Agronomy, Faculty of Nature and Life Sciences, University of El Oued, El Oued 39000, Algeria; 8Nuclear Research Centre of Birine, Ain Oussera, Djelfa 17200, Algeria; 9Department of Plant Production Technology and Commodities Science, University of Life Science in Lublin, 20-950 Lublin, Poland; 10Scientific Consulting, Chemical Engineering, University of Chemical Technology and Metallurgy, 1734 Sofia, Bulgaria

**Keywords:** phenolic compounds, medicinal plants, traditional healers, ethnobotany, illnesses, climatic conditions

## Abstract

Due to their robust antioxidant capabilities, potential health benefits, wide variety of biological activities, and strong antioxidant qualities, phenolic compounds are substances that have drawn considerable attention in recent years. The main goal of the review is to draw attention to saharian Algerian medicinal plants and the determination of their bioactivity (antioxidant, anti-cancer, and anti-inflammatory importance), and to present their chemical composition as well as in vivo and in vitro studies, clinical studies, and other studies confirming their real impact on human health. Research results have revealed a rich variety of medicinal plants used to treat various disease states in this region. Based on in vivo and in vitro studies, biological activity, and clinical studies, a list of 34 species of desert plants, belonging to 20 botanical families, useful both in preventive actions and in the treatment of neoplastic diseases has been established, and polyphenolic compounds have been identified as key to the health potential of endemic diseases and desert plants. It has been shown that people who follow a diet rich in polyphenols are less prone to the risk of many cancers and chronic diseases, such as obesity and diabetes. In view of the increasing antioxidant potential of these plant species, as well as the increasing trade in herbal products from the Sahara region, phytosanitary and pharmaceutical regulations must change in this respect and should be in line with Trade Related Aspects of Intellectual Property Rights (TRIPS), and the sustainable use and development of plant products must be addressed at the same time.

## 1. Introduction

Recently, a great deal of research has been completed on traditional medicine and its applications. It has been determined that conventional medicine is a body of knowledge, abilities, and procedures based on concepts, beliefs, and experiences that are particular to various cultures and are used to maintain health as well as prevent, identify, treat, and improve a variety of physical and mental illnesses [1,2,3].

On the other hand, the study of medicinal plants is a method for examining the links and interactions between the biological and cultural components of the environment [4]. Nowadays, ethnobotanical studies are seen to be the best method for finding new medicinal plants or concentrating on those that have already been recognized as having bioactive components [5], and only a few studies have been completed in the Hot Arid Regions in terms of assessing chemical contents of medicinal plants, especially in terms of identifying the structure of bioactive elements of traditional medicinal plants [6].

The Sahara Desert encompasses the majority of the northern part of the African continent, stretching from the Atlantic to the Red Sea, and because of the extremely low and irregular precipitation, high temperatures, wide temperature range, and protracted droughts that characterize this eco-region’s climate, many living organisms find it difficult to survive [7]. Furthermore, this eco-region’s geomorphological characteristics are primarily related to the growth of sporadic vegetation [8]. Although there is relatively little rainfall each year, it is enough to support plant life on practically all northern Sahara Desert landscapes. Thus, a landscape-specific vegetation has an extremely unbalanced density and structure that are influenced by the habitat’s environmental factors. In depressions such as Wadi beds, the flora is thicker, but on desert plateaus, pavements, and sand dunes, there is scattered vegetation with limited canopy cover [9].

Under these harsh climatic conditions and a variety of pressures in these desert areas, however, many Saharian plants have been discovered that are still in use today as they have developed thanks to ethnic medicine, including morphine, opium, and anesthetic alkaloids, these plants may act as storage for natural, safe, and effective macromolecules that can be used as antioxidants [10,11,12].

The aim of this review was to discover plants that grow spontaneously in the Saharian and Algerian areas (Hot Arid Regions) that are utilized in traditional medicines by indigenous people. This is the first attempt at a comprehensive study of the therapeutic qualities of such medicinal plants, which are likely to attract pharmacologists and biological control researchers for further critical and scientific confirmation.

## 2. Methodology

### 2.1. Collection of Information

Published literature between 2000 and 2021 in the form of books and articles downloaded from databases: PubMed, Scopus, Science Direct, Wiley Online Library, and Google Scholar using keywords such as medicine practitioner, traditional medicine, and traditional medicinal plant were used as sources of information.

### 2.2. Identification of Medicinal Plant

A comprehensive ethnobotanical survey of medicinal plants in Saharian Algerian areas was conducted to gather information on the most significant families of ethnomedicinal plant species (Appendix A) utilized by the inhabitants of the research area. Plant materials were identified with the assistance of humans and healers [13,14].

## 3. WHO’s View of Traditional Medicine

According to the World Health Organization (WHO), traditional medicine is used by 80% of the people in the developing world. In recent decades, the industrialized world has seen an increase in the use of complementary and alternative medicine (CAM), notably herbal treatments [15]. Herbal medications are made up of herbs, herbal materials, herbal preparations, and completed herbal products that have active components that are plant parts or other plant materials [16]. While herbal medicines are used by 90% of the population in Ethiopia for basic healthcare, studies in affluent nations such as Germany and Canada reveal that at least 70% of the population has tried medicine (CAM) at least once [2,15]. It’s possible that ancient civilizations’ extensive knowledge of herbal medicines, established through trial and error over many years, as well as the most significant cures, were carefully passed down from generation to generation [2].

Indeed, contemporary allopathic medicine has its roots in this old medicine, and many significant novel treatments are expected to be created and marketed in the future from African biodiversity, just as they have in the past by following the leads offered by traditional knowledge and experiences [17,18].

## 4. Polyphenolic Compounds and Bioactivity

The polyphenol profile of alcoholic or aqueous extracts of medicinal plants has recently been the subject of various investigations. For instance, the paper of Binello et al. [19,20] demonstrated microwave-assisted extraction (MAE) and ultrasound procedures for the selective green extraction of polyphenols from lemon balm, where it was found that the compound rosmarinic acid is the main constituent of the phenolic fractions, and it was also determined that ethanol is an excellent solvent for both MAE ultrasonography methods.

Reactive oxygen species are involved in a variety of illnesses; hence, plant extracts containing low molecular mass chemicals have been utilized in phytotherapy since ancient times. Many naturally occurring substances have been shown to have significant action as radical scavengers and lipid oxidation inhibitors [19,20].

Plant antioxidants such as phenolic compounds (see Figure 1) (tannins, flavonoids, anthrocyanins, chalcones, xanthones, liganans, depsides, and depsidones), terpenes (sesquterpens and diterpines), alkaloids, and organic sulfur compounds, in addition to alpha-tocopherol and beta-carotene, are useful as antioxidants [21]. A significant number of studies on the antioxidant activity of various plant extracts and powders have been conducted. The findings of these tests show that many secondary metabolites, particularly phenolic compounds such as flavonoids and tannins, etc. [22].

### 4.1. Polyphenols

Numerous plants contain polyphenols, which are vital nutrients in the diets of both people and animals. The finest polypharmacy against the onset of chronic illness is found in fruits and vegetables, which include a wide range of antioxidant components, such as polyphenols [23]. Polyphenols play an important role in the scavenging of free radicals. In several herbs, vegetables, and fruits, there was a clear link between antioxidant activity and total phenolic concentration [24].

Polyphenols are a sizable class of physiologically useful chemicals that are found in nature; they may be broken down into four categories [25]. Bioflavonoids are the first group. Anthocyanins and proanthocyanidins are two families of bioflavonoids that are closely related (OPCs). The xanthones are the final group.

### 4.2. Flavonoids

Flavonoids work as antioxidants by neutralizing oxidizing radicals such as superoxide and hydroxyl radicals, as well as through chelation [26]. Due to the electron-donating capability of their phenolic groups, flavonoids can also serve as a strong chain-breaking antioxidant. Flavonoids have powerful antioxidant activity; their capacity to scavenge hydroxyl radicals may be their most significant function, and it underpins many of their activities in the body [27].

Flavonoids have been shown to protect against ischemia tissue damage by serving as free radical scavengers, and by functioning as antioxidants, they have numerous positive benefits such as anti-inflammatory, antiallergic, antiviral, and anticancer properties. They’ve also been linked to a reduction in the risk of liver disease, cataracts, and cardiovascular disease [28]. They have direct antioxidant action as well as protective effects on other antioxidants such as vitamins C and E [29]. In addition, their ability to influence membrane-dependent processes such as free radical-induced membrane lipid peroxidation is linked not only to structural features but also to their ability to interact with and permeate lipid bilayers [30].

Polyphenolic compounds are especially susceptible to peroxynitrite-dependent reactions and are powerful inhibitors of nitrous acid-dependent nitration and DNA deamination in vitro, and the role can be exerted in vivo; thus, flavonoids may provide gastro-protective effects when high levels of RNS are produced, in addition to their effects on ROS [31].

Anthocyanins are a kind of flavonoid, which is a family of antioxidant chemicals. Anthocyanins are the pigments that give red, purple, and blue plants their vibrant colors. They may be found in a variety of foods. Anthocyanins may have anti-inflammatory, antiviral, and anti-cancer properties in addition to serving as antioxidants and combating free radicals [32,33].

Anthocyanin-rich compounds have long been utilized in herbal medicine to treat a variety of blood vessel-related diseases, including chronic venous insufficiency, high blood pressure, and diabetic retinopathy. They have also been used to treat a variety of other illnesses, such as colds and urinary tract infections. Anthocyanins may also help protect against major health issues such as heart disease and cancer [34].

### 4.3. Alkaloids

A diverse set of naturally occurring compounds known as alkaloids perform a wide range of biological functions. Many of these have important medical uses. The most well-known alkaloids, such as morphine, quinine, strychnine, and cocaine, are made from plants, which are also the first source of alkaloids [35]. The identification of novel alkaloids in these less complex microbial species has benefited from the quick advances in molecular biology and genome sequencing, and a wealth of biosynthetic knowledge about these substances has been acquired in a number of recent mechanistic studies. There are certainly still a lot more microbe-derived alkaloids to be discovered [36]. Metabolic engineering initiatives are a significant biosynthetic research application since many alkaloids have medicinal utility. Alkaloid production has been increased by metabolic engineering, and the biosynthetic enzyme engineering of alkaloids has been used to logically alter the structure of alkaloids in simpler hosts. Recently, there has been substantial advancement in the study of alkaloids, biosynthetic pathways, and metabolic engineering [37,38,39,40].

## 5. Discussion

### 5.1. Search Results

The main emphasis in the current review was on the most important types of plants used (Table 1). From the search above, more than 150 ethnobotanical articles and phytochemistry and pharmacology papers were retained and have been approved by this review.

### 5.2. Ethnobotanical Studies

Table 1 shows a list of medicinal plants used in Saharan regions for traditional treatment, as well as their habits, portions utilized, route of administration, nations where they are used, toxicity, and ethnopharmacology. Using websites, the proper names and authorities were listed according to the International Convention of Botanical Nomenclature ((www.theplantlist.org, accessed on 2 January 2023), JSTOR (http://plants.jstor.org, accessed on 2 January 2023), and Tropicos (www.tropicos.org, accessed on 2 January 2023))

### 5.3. Pharmacological Studies

Several plants have demonstrated promising results in studies. In vitro, in vivo, and in clinical trials, it possesses healing effects. We provide here pharmacological research, summarized in Appendix A, that has looked into Saharan medicinal plants that are used to cure a variety of illnesses directly or indirectly. Some of these plants were not found in ethnobotanical studies, and the author is the only one who can confirm their usage in traditional medicine.

### 5.4. Ethnopharmacology and Toxicological Evidence

Due to the lack of dose recommendations in herbal medicine, additional drug discovery methods such as toxicological, clinical, and lab studies are required to scientifically confirm folkloric use and unravel the possible toxicity of the implicated plants [39]. Through this study, not enough information has been found about the toxicity of these plants. This implies that more work still needs to be completed on medicinal plants before they are recommended to patients to prevent adverse side effects.

### 5.5. List of Multi-Purpose Plants and Their Uses in the Saharan Regions

#### 5.5.1. *Retama retam Webb.* (Family: Fabaceae; Vernacular Name: Rtem)

*Useful parts*: Aerial part (Infusion, powder, compressed herbal).

*Investigation:* Analgesic, antiseptic, and anti-inflammatory.

The stem of this plant is utilized in cauterization in traditional medicine. It can also help with rheumatism, scorpion stings, and injuries. Despite their obvious relevance in traditional medicine, the number of studies on chemical components with biological activities (therapeutic virtues) on the *Retama retam Webb.*, particularly in Algeria, is sparse compared with other herbs [40].

Selaimia A et al., (2020) assessed antioxidant activity using the DPPH technique. The findings of the DPPH activities indicated that the extracts of the *Retama* had the most antioxidant activity, with the ethyl acetate extract of the leaves having the highest antioxidant activity. According to the antimicrobial activity research, the extracts have a varied activity that varies depending on the strain examined. It has been found to be ineffective against meticillin-resistant Staphylococcus aureus, although it is extremely sensitive to Candida albican [40].

Another investigation found that the extracts of these plants, dried in various ways, exhibit antibacterial activity suppression for both bacterial strains (*Escherichia coli* and *Pseudomonas aeruginosa*). *Retama retam*, on the other hand, was shown to have limited antibacterial action against *Pseudomonas aeruginosa* and high antibacterial activity against other bacteria [41,42]. These findings suggest that the herb might be useful in the treatment of viral infections. Their ability to distinguish between different microbes.

#### 5.5.2. *Astragalus cruciatus Link* (Family: Fabaceae; Vernacular Name: Akifa)

*Astragalus cruciatus Link.* belongs to the genus Astragalus L. of the family Fabaceae. Ithas four synonyms “*A. aristidis Coss*.”, “*A. radiatus Ehrenb.*”, “*A. trabutianus Batt.*”, and *A. corrugates Bertol* [43,44].

Species of the Astragalus genus are used as therapeutic herbs in traditional medicine all throughout the world to treat stomach ulcers, cough, chronic bronchitis, hypertension, gynecological problems, diabetes, and scorpion poisonous stings [45]. Immunostimulant, cardiovascular, and antiviral properties have been found in plants from the same genus [46,47]. Saponin, phenolic, and polysaccharide chemicals are the physiologically active ingredients of Astragalus species, while thenitro-toxins, imidazoline alkaloids, and selenium derivatives are poisonous components [48].

The Astragalus genus is well-known for its abundance of bioactive secondary metabolites. Saponins [49,50,51,52,53,54] and flavonoids [55,56,57] have been isolated and characterized in previous phytochemical studies of several Astragalus species. However, there has never been any constitutional or pharmacological research on *Astragalus cruciatus Link.*

#### 5.5.3. *Genista saharae* (Coss. & Dur.) (Family: Fabaceae; Vernacular Name: Marekh)

*Investigation:* Cold, influenza, respiratory system problems. It contains flavonoids compounds [49].

*Genista saharae* is a leafless, spontaneous fabaceae that is colloquially called as “Tellegit” in Algeria. For the reason that it is known as a source of chemical components with antioxidant properties, this medicinal plant is used in traditional pharmacopeia.

Sofiane Guettaf’s research focuses on examining the phytochemical content and assessing the antioxidant activity of aerial portions of an aqueous extract of *Genista saharae* (AEG) under settings similar to its traditional use. The qualitative chemical composition of AEG was assessed via phytochemical screening utilizing precipitation and coloring reactions. Furthermore, spectrophotometric techniques were used to determine the total phenolics, flavonoids, tannins, and carotene contents. Finally, the antioxidant properties of AEG were determined using three different methods: DPPH, reducing power, and β carotene bleaching tests. Phénolic compounds, flavonoids, alkaloids, tannins, terpenoids, glycosides, steroids, and saponins are among the biomolecules found in AEG, according to the findings. Furthermore, the quantitative examination reveals a significant amount of total phenolics, tannins, and β carotene. In sum, EAG has a strong antioxidant activity against β carotene bleaching, which validates its usage by traditional healers. As a result, *Genista saharae* is an excellent antioxidant source. The presence of tannins or other phenolic chemicals such as terpenoids, β carotene, and saponins may explain its high antioxidant activity [40].

The chemical composition and antioxidant activity of this genus’ alcoholic extracts have been studied before [51,52,53]. Unfortunately, there is presently no phytochemical or biological information regarding this shrub. There is no information in the literature on the antioxidant effects of its aqueous extract. As a result, all of the *Genista saharae* findings are consistent with its traditional use.

#### 5.5.4. *Astragalus gyzensis Bunge* (Family: Fabaceae; Vernacular Name: Foul Alibil)

*Investigation:* Scorpion stings and snake bites. According to ref. [54], this plant is used to treat snake bites and is a highly hardy plant in the Saharan climate, which is characterized by a variety of stresses. Animals find this plant to be moderately appealing, especially when it is in blossom. Volatiles, which are constantly released into the atmosphere, can account for this. Many studies involving the essential oils of plants have been conducted in recent years for the purpose of discovering natural products and anti-disease actives. Essential oils contain a variety of biological activities, including larvicidal efficacy against mosquitos [55,56]. The essential oil may be utilized to produce natural pesticides and help reduce the negative effects of synthetic goods, including buildup, resistance, and contamination.

#### 5.5.5. *Euphorbia guyoniana Boiss. & Reut* (Family: Euphorbiaceae; Vernacular Name: Lebina)

*Investigation:* Diarrhea, skin diseases, scorpion stings and snake bites.

This plant, similar to many euphorbias, is extremely poisonous due to the presence of toxic white latex. The nomads, on the other hand, utilize it to protect themselves from snake bites [54]. *Euphorbia guyoniana Boiss.* and *Reut.*, a member of the Euphorbiaceae family, was the plant studied in this study. This species may be found across Algeria’s desert and pre-desert areas [13,57]. It’s called “Lebbina” in the area and grows 30–100 cm tall with upright and branching stems that produce very poisonous white latex. More than 2000 species of Euphorbia have been used in traditional medicine to treat skin problems, including dermatitis, gastrointestinal issues, bacterial or fungal infections, and even some forms of cancer. Algerian Sahara nomads also utilize *E. guyoniana* to protect themselves against snake bites [58]. It has significant antioxidant capabilities, according to a recent study [59]. The high amount of secondary chemicals in *E. guyoniana*, such as terpenoids, alkaloids, and flavonoids, contributes to its positive benefits.

#### 5.5.6. *Ephedra alata DC.* (Family: Ephedraceae; Vernacular Name: Alanda)

*Parts of use:* Leaves and boughs (Maceration, inhalation, herb tea).

*Investigation:* This plant is used to cure a variety of ailments, including colds, influenza, respiratory issues, hypertension, body aches, whooping cough, and cancer [13,49,60,61].

#### 5.5.7. *Heliathemum lipii* (L.) *Pers*. (Family: Cistaceae; Vernacular Name: Samhari)

The genus *Helianthemum* belongs to the family Cistaceae [62], which are widespread in the Mediterranean region. Across the world, this genus includes 70 species [63]. In Algeria and Pakistan, this genus has a single species *Helianthemum lippii* (L.) *Pers.*

The vernacular name of this species is different in different regions and continents, for examplef the name of Al Samhari (in the region of Oued Souf: Southeast of Algeria) [49]; Reguig (in the region of Ouargla: South of Algeria) Tahsowat and Alrjik (South-West of Algeria); Alrkaroq (in Kuwait); Umm Souika (the Arabian Peninsula) [64]; and Sun Flower (Northeast of Jordan) [65].

This plant is highly intriguing from an ecological and economic standpoint [66]. It belongs to the pastoral plants [49,66], plays a key role in the battle against desertification, and contributes to the stability of desertification-prone areas. The species’ powder is used to treat skin rashes [66]. It’s also used in Libya to treat rot and rashes, as well as to prevent illness [67,68]. In Morocco, this plant with the Bedouins can cause lameness in camels in the name of GAF or Kraft (a type of arthritis). But in reality, the toxicity of this plant has not yet been proven [69].

#### 5.5.8. *Cyperus conglomeratus* (Family: Cyperaceae; Vernacular Name: Saad)

*Cyperus rotundus* is one of the best known species of the genus as a medicinal plant, featuring in Indian, Chinese, and Japanese traditional medicines for spasms, stomach and intestinal problems, and menstrual irregularities [70]. Several writers have studied *C. rotundus* extensively; the most notable essential oils isolated from it are α-pinene, β-pinene, α-copaene, cyperene, cyperotundone, α-cyperone, and caryophyllene oxide [71,72,73,74,75,76,77]. For the reason that the southern coast, eastern south, and central parts of Iran have dry and very hot air for eight months of the year, this species can withstand harsh environmental conditions [78]. This species has been recorded to be used as a pectoral, emollient, diuretic, stimulant, analgesic, and anthelmintic therapy in traditional medicine.

#### 5.5.9. *Calligonum comosum L’herit* (Family: Polygonaceae; Vernacular Name: Larta)

*Parts of use:* Leaves, roots, boughs (infusion, decoction).

*Investigation:* This plant can be used against scorpion stingsand snake bites [13,60], vermifuge.

The antibacterial activity of *Calligonum comosum L. Her.*, “Arta,” a member of the family Polygonaceae, was investigated. It is a tropical and subtropical plant that is widely distributed in the United Arab Emirates and Saudi Arabia. Previous research found that an ethanolic extract of *Calligonum comosum* aerial parts greatly decreased the rise in carrageenan-induced hind paw oedema in rats. Furthermore, a pre-treatment with the extract inhibited the acute stomach ulcers caused by phenylbutazone and indomethacin in a dose-dependent manner [71]. Recently, researchers discovered that extracts from several *Calligonum comosum* plant sections have significant antibacterial activity against four harmful bacteria [72]. Anthraquinones and flanovonids are the most frequent chemical components of Arta, according to prior investigations. In addition, when *Calligonum comosum* was treated with organic solvents, dehydrodicatechin was extracted, which showed cytotoxic and antioxidant activities [73,74].

#### 5.5.10. *Plantago albicans* L. (Family: Plantaginaceae; Vernacular Name: Linem)

Is an herbaceous plant that grows wild in subtropical and temperate climates and may be grown readily in Tunisia, Algeria, and Libya. The extracts of Plantago species are often used in traditional medicine due to their hepatoprotective [75], analgesic, anti-inflammatory, and antipyretic properties. Furthermore, *P. albicans* caused structural and functional changes in liver and heart tissue [76]. In fact, this genus contains a high amount of primary and secondary metabolites [77].

#### 5.5.11. *Limoniastrum guyonianum Coss & Dur.* (Family: Plumbaginaceae; Vernacular Name: Zita)

*Investigation:* This plant can be used as a treatment for several diseases, including: diabetes, scorpion stings and snake bites, headaches, constipation, hypertension, renal illness, anemia, antiseptics, burns, leprosy, wounds, ulcers, diabetes, anemia, cough, constipation, gas, obesity, tonsillitis, and flu [78].

#### 5.5.12. *Tamarix boveana* (Family: Tamaricaceae; Vernacular Name: Tarfa)

*Investigation:* This plant can be used as a treatment for several diseases, including: cough, hemorrhage, diuretic, antiseptic, leprosy, injuries and ulcers, scorpions and insect bites, renal diseases, diarrhea, anemia, gum and mouth inflammation, gastric ulcer, cephalalgia, hypertension, diabetes, joint disease, and pancreatic inflammation [78].

#### 5.5.13. *Traganum nudatum Del.* (Family: Chenopodiaceae; Vernacular Name: Damran)

*Parts of use:* Leaves (compressed maceration, powder, and ointment).

*Investigation:* Rheumatism, skin diseases, diarrhea, rheumatism wound dermatoses [54]. This plant is well renowned for its tasty fruit as well as its wood for burning [60].

#### 5.5.14. *Bassia muricata* L. Asch. (Family: Chenopodiaceae; Vernacular Name: Ghabitha)

*Investigation:* Analgesic, antiseptic, and anti-inflammatory. It is a rich plant with triterpenoids and saponins [49].

It grows in sandy environments in North Africa, the East Mediterranean region, Sinai, Saudi Arabia, and Iran as an annual plant [79]. It is a significant plant in traditional medicine, where it is used as an analgesic, antipyretic, and nephrotic. It also contains various biological properties, including antioxidants [80]. The extract of *B. muricata* has been shown to lower white blood cell counts, enhance prothrombin time, reduce blood pressure, and have antibacterial properties [81]. Flavonoid glycosides and other phytochemical components of *B. muricata* have been isolated and identified [82,83,84].

#### 5.5.15. *Atriplex halimus* L. (Family: Chenopodiaceae; Vernacular Name: Gatef)

*Investigation:* this plant can be used as a treatment for several diseases, including: cysts in the uterus, diabetes, stomach pain, constipation, diarrhea, gas, hypertension, antiseptic, burns, fever, anemia, otitis, rheumatism, diuretic, vermifuge, vomiting, wounds and ulcers, tonsillitis, gallbladder disease, fortify the gums, infertility, prostate, fall of placenta, nephrolithiasis, hypercholesterolemia [78,85].

Streptozotocin-induced diabetic rats showed both long-term and short-term antidiabetic effects. The aqueous extract was given orally at a dose of 200 mg/kg BW for 30 days to rats divided into four groups: normal rats, diabetic rats (50 mg/kg BW of streptozotocin), and diabetic and normal rats treated with 200 mg/kg BW in the short-term study after administration of a glucose dose of 3 g/kg [86]. When compared with the results before treatment, the aqueous extract resulted in a substantial reduction (54%) in blood glucose in the treated diabetic group at the end of the trial. The plant extract also has an antihyperglycemic effect, lowering diabetic rats’ fasting blood glucose levels by 23 and 41% after 2 and 3 h, respectively [86]. The aqueous extract is high in tannins and flavonoids, which the researchers believe may have an effect on the pancreas via increasing beta cell insulin production.

#### 5.5.16. *Zygophyllum album* L. (Family: Zygophyllaceae; Vernacular Name: Agga)

*Parts of use:* Leaves, stems, and fruits (Decoction, powder, ointment).

*Investigation*: Diabetes, purgatives and laxatives, anti-viruses and fungi, indigestion.

This plant can be used to treat diabetes, indigestion, skin problems, as an analgesic, and as a disinfectant, according to [60]. This herb is utilized in Tunisian traditional medicine as a rheumatism, gout, and asthma treatment [87]. It’s also a diuretic, a local anesthetic, an antihistaminic, and an antidiabetic [88].

#### 5.5.17. *Matricaria pubescens* (Desf) (Family: Asteraceae; Vernacular Name: Guartoufa)

*Parts of use:* Leaves (Powder).

*Investigation:* It is used as a treatment for: scorpion stings and snake bites, colds and respiratory difficulties, bleeding, diuretics, fever, astringents and stimulants, stomach and stomach aches, and constipation. This herb is used to treat scorpion stings and snake bites in the Oued Righ area (El Oued, Algeria) [54].

#### 5.5.18. *Launaea resedifolia O. K.* (Family: Asteraceae; Vernacular Name: Athid)

The *Launaea resedifolia O. K* family is one of the biggest angiosperm families, with around 340 genera and 3350 species divided into 11 tribes. They are found all over the world, primarily in temperate parts of the northern hemisphere [89]. The Irano-Turanian, Mediterranean, and Saharao-Sindian areas are the family’s primary distribution hubs.

#### 5.5.19. *Solanum nigrum* L. (Family: Solanaceae; Vernacular Name: Anb Aldib)

*Parts of use:* Leaf, stem, and fruits are the parts of the plant that are used. It is a poisonous plant that is classified as active and hazardous in the pharmacopoeia. It is for external use [54].

*Investigation:* It is used as a treatment for: diuretic, chronic liver enlargement, diarrhea, and piles; also effective against skin illness and anthrax. Fruits are used to treat a variety of ailments, including heart disease, hiccups, asthma, fever, bronchitis, and diarrhea. Ringworm can be treated using green fruit pastes. Fruit juice is an expectorant and cooling drink that can be used to treat fevers, gonorrhea, giddiness, and inflammations [90].

#### 5.5.20. *Erodium glaucophyllum* L. *Her.* (Family: Geraniaceae; Vernacular Name: Tommir)

*Parts of use:* All its parts are useful.

*Investigation:* It is used as a treatment for: diarrhea, colds, influenza, and respiratory system disorders are investigated. It’s a medicinal herb that’s good for diarrhea, an astringent, allergies, and oxytocin [54]. *Erodium glaucophyllum* L. is a common herb in the Nile valley, the western Mediterranean coastal region, and the deserts, and goes by the Arabic names Kahkul, Lesan Hamad, Kabshia, Ragma, Dahma, Murrar, and Tamir.

#### 5.5.21. *Cleome arabica* L. (Family: Capparidaceae; Vernacular Name: Nettin)

*Parts of use:* Leaves (Infusion, maceration).

*Investigation:* Rheumatism, urinary tract.

It is a plant that is high in flavone chemicals, particularly flavonoids, and is diuretic. It can also help with rhumatism, arthritis, and diarrhea [13,49,60].

#### 5.5.22. *Neurada procumbens* L. (Family: Rosaceae; Vernacular Name: Anfal/Saadan)

*Parts of use*: Leaves, seeds and fruits.

*Investigation:* Analgesic, antiseptic, anti-inflammatory, astringent, and stimulant. The Bedouin have always regarded them as edible medical plants [91]. The Bedouins see it as a sort of camel meal that is both safe and edible. The herb has also been used to cure diarrhea and dysentery in traditional Arabic medicine. It’s also been utilized to boost heart and respiratory processes as a stimulant [91]. Its water extract made the mice’s blood pressure rise. Only one study examined the chemical constituents of this plant, which revealed that it contains alkaloids, flavonoids, saponins, sterols and terpenes, volatile oils, and tannins, as well as a hypertensive effect of its ethanol extract, indicating that cardiovascular patients should exercise caution when using this plant [92]. It was also used to cure stroke in people and animals in Pakistan, and its dried star-shaped fruits may be mixed with rose water in the summer as a cooling agent and with dry nuts in the winter to stimulate nerves [92]. In addition to dihydroflavonol glycosides, prior findings in Egypt suggest that this plant contains polysaccharides (gum, mucilage) [92]. On the other hand, mineral monitoring, particularly for toxic elements, is one of the most important aspects of ensuring the quality of medicinal plants because elements such as Fe, Ca, and Zn play an important role in many processes of human metabolism and thus contribute significantly to human health [93,94,95,96], as well as plant growth processes, even in small amounts.

## 6. General Debate

There are many plants with therapeutic potential, as has become evident over the past few decades. It is also commonly recognised that traditional medicinal plants may be able to provide prospective template molecules for the development of new drugs. Numerous of the plants included here show very potential therapeutic properties, suggesting that more clinical research should be completed on them. However, only a few of them have compelling scientific and clinical support [97,98,99]. Considerable work should be put into finding and describing the bioactive components present in these plants, given the state of scientific understanding [100]. It is a challenging and potentially sophisticated medical process to identify therapeutic components in traditional plant-based remedies [101]. There is still a dearth of literature resulting from the last decade’s investigations addressing procedures to be adopted for quality assurance, authentication, and standardization of crude plant products, despite ongoing thorough and mechanism-oriented evaluations of medicinal plants from the Saharan region’s flora. The formulation of the finished product, extraction techniques, and proper raw material management may all contribute to achieving the desired consistency [16,102]. In fact, it has been acknowledged that one of the main obstacles to the formation of a modern phytomedicine business in the Saharan region is the lack of proper validation of traditional knowledge, as well as technological needs and quality control standards. For buyers, both domestic and foreign, this makes assessing the efficacy and safety of plants and extracts, as well as contrasting batches of commodities from different locations or years, extremely challenging [102,103]. Potential safety issues, such as the contamination of medicinal plant products with heavy metals from traditional medical supplies from the Saharan region, must also be addressed, and regulatory regulations must be properly developed and put into place. By using controlled settings for growth (under GACP) and processing, contamination of medicinal plant material must be kept to a minimum (under Good Manufacturing Practice). For the reason that it is simpler to regulate the supply chain and contamination is limited, cultivated plant material is chosen in the medicinal plant sector [104,105].

On the other hand, correct identification of medicinal plant material is critical to the quality control process; the source of the plant material must be established unambiguously. Following that, during the material’s processing phases, microbiological contamination (fungal, bacterial, and any possible human diseases) must be verified. Chemical, pharmacological, and toxicological assessments, completed in accordance with Good Laboratory Practices (GLPs) principles, will validate the bioactive characteristics of the material during processing [106].

These tests are routinely used to predict the safety of newly produced products, where clinical safety and efficacy must be established in lengthy, in-depth investigations during the early stages of a medicinal agent’s development. The unit dosage forms generated after that will be regarded as safe as long as the standard operating procedures are followed. Regardless, quality assurance processes must be implemented to ensure that the factory’s goods are of excellent quality, safety, and efficacy [107].

Alkaloids and flavonoids have well-known antimicrobial and spasmolytic properties [108]. In contrast to alkaloids, catechic and gallic tannins, flavonoids, and saponins are abundant in organic extracts of *P. lentiscus* leaves, according to research by Barbouchi [109]. The A. herba-alba is abundant in compound phenolics, flavonoids, tannins, and anthocyanins, as shown by Khlifi’s study [110]. According to Najafi et al. [111], the ethanolic extract of *C. colocynth* seeds includes tannins, alkaloids, flavonoids, and saponins, whereas Benariba et al. [112] found that catechic tannins and flavonoids are plentiful in hydromethanol extracts. These variations in outcomes may be related to variations in the harvesting region, soil composition, climate, harvest season, solvents, and experimental extraction settings.

## 7. Health Claims Based on Biological and/or Therapeutic Activities

The phenolic acid derivatives and polyphenolic components, primarily flavonoid glycosides, are the primary biological components of most of the studied therapeutic plants and herbs [113]. Flavonoids are so-called secondary plant chemicals that have a variety of physiological and pharmacological effects (see Figure 2) [114]. They possess diverse biological properties such as antioxidant, antiageing, anti-carcinogen, anti-inflammatory, anti-atherosclerosis, cardioprotective and improved endothelial function [115]. The majority of these biological effects are thought to be due to their inherent decreasing capacities. They may also provide indirect protection by triggering endogenous defense mechanisms and altering physiological processes [116].

Phytosterols are another category of chemicals found in these plants and herbs. The most notable of their bioactivities is their ability to decrease blood cholesterol by partially inhibiting intestinal cholesterol absorption [117]. Possible antiatherogenic action is one of the claimed benefits of phytosterols, as well as the immune boosting and anti-inflammatory activities carried out mostly by beta-sitosterol, which are all claimed advantages of phytosterols. Furthermore, there is growing evidence that certain of these plants and herbs’ sterols may have especially protective benefits against the development of various malignancies, including colorectal, prostate, and breast cancers [118,119]. It’s unclear if mechanisms other than phytosterols’ well-known cholesterol-lowering effect may also play a role in these possible health advantages [120].

### 7.1. Antioxidant and Detoxicating Activity

A molecule known as an antioxidant is one that can slow down or prevent the oxidation of other molecules. In the chemical process of oxidation, electrons are moved from a substance to an oxidizing agent [121]. Oxidation activities generate free radicals, which trigger a cascade of detrimental cell-damaging events. Antioxidants can work as free radical scavengers, helping to eliminate dangerous free radicals and their byproducts while also inhibiting further oxidation reactions by being oxidized. Several factors, some of which are depicted in Figure 3, influence how effective antioxidants are. Antioxidants such as thiols or polyphenols are therefore commonly utilized as reducing agents [122,123]. According to the radical hypothesis of human physiology, active free radicals are involved in almost every cellular breakdown process that results in cell death. Oxidative stress is thought to play a role in a number of chronic and degenerative diseases, including cancer, autoimmune disorders, aging, cataracts, rheumatoid arthritis, cardiovascular diseases, and neurological disorders [124]. In the majority of the plants analyzed, phenolic components have been linked in several studies to a significant degree of antioxidant bioactivity. The relationship between these two characteristics, however, is not entirely evident [125,126].

### 7.2. Anti-Inflammatory Activity

Inflammation is a physiological response to physical or biological substances causing harm to tissues or cells, comprising a variety of responses aimed at removing the source and repairing the damage. Polyphenols are one of the most common types of phytochemicals with anti-inflammatory characteristics, and many plants with polyphenols as secondary metabolites have been shown to have potential anti-inflammatory capabilities [127].

### 7.3. Benign Prostatic Hyperplasia

A burning sensation, a strong and frequent urge to urinate, discomfort in the lower back or abdomen, and difficulty urinating can all be symptoms of prostatitis or prostate inflammation. The most important use of plants or herbs in medicine is for the preventative and curative treatment of prostate disorders. An enlarged prostate is known as benign prostatic hyperplasia (BPH). BPH is caused by non-cancerous growths inside the prostate. Chronic prostatitis is particularly frequent in senior men, possibly due to hormonal changes and aging. It is not surprising that more people are turning to phytotherapy and other alternative treatments as opposed to pharmaceutical therapies because they have fewer side effects than antibiotics.

### 7.4. Antidiabetic Activity

Aqueous extracts of leaves from numerous plants have been found to help reduce blood glucose levels. Additionally, it exhibited a sizable hypolipidemic effect, bringing down blood triglyceride and total cholesterol levels. The plant extracts outperformed the synthetic drug metformin significantly in terms of antihyperglycemia and antihypertriacyl-glycerolaemia. The results also suggest that the plant extract could be used to treat type 2 diabetes and the associated dyslipidemia. These polyphenolic compounds act as monomers or oligomers in epididymal fat cells, enhancing insulin activity in vitro and demonstrating insulin-like action as well as an antioxidant impact in vitro [128].

### 7.5. Antiviral Effect

For the reason that viruses have defied prevention or therapy longer than any other form of life, infectious viral illnesses continue to be a serious danger to public health. Traditional folk medicine utilizes medicinal plants to treat a variety of illnesses, including infectious infections. At 1000 g/mL with Rf 104, hydro-alcoholic extracts of several medicinal plants were shown to have virucidal action against the herpes simplex-1 virus (HSV). The capacity of plant extract dilutions to suppress the generated cytopathogenic effect (CPE) is represented as a reduction factor (Rf) of the virus titer in an antiviral bioassay. The study was carried out by the University of Bristol in the UK using a technique called end point titration technique [127].

### 7.6. Antineoplastic Properties

Only evaluating the effect on cell viability, as some of the reports included in our review (Appendix A) did, is not a valuable report because cell cultures are highly susceptible to environmental factors such as temperature and osmotic pressure. Another reason for the low reliability of in vitro antineoplastic studies is that in a living organism, the plant extract is exposed to several other cells besides the target tissue; thus, the extract may be toxic to normal tissues as well, which is one of the main reasons why several agents have been barred from clinical use as anticancer agents. Another issue is that malignant cells generally acquire resistance to anticancer medicines after prolonged exposure, which is one of the main reasons why researchers are continuously looking for novel anticancer treatments. As a result, in vitro antineoplastic activity does not imply anticancer activity, and more data, such as antitumor characteristics in tumor-bearing animals, is needed to determine whether a plant is worthy of further investigation as a source of anticancer drugs [128].

## 8. Traditional Medicine

Medicinal plants used to cure humans have bestowed a plethora of herbal remedies on the Sahara, which local people acquire, maintain, and pass on to the next generation [129]. The widespread usage of traditional medicine in Hot Arid Regions, which is mostly comprised of medicinal plants, has been connected to cultural and economic factors. As a result, the World Health Organization (WHO) urges member desert nations to promote and incorporate traditional medicinal practices into their health systems [1]. Plants generally include a variety of phytochemicals, also known as secondary metabolites, that can benefit health singly, additively, or synergistically [130].

Indeed, unlike pharmacological medicines, medicinal plants frequently have multiple compounds acting together catalytically and synergistically to generate a combined impact that is greater than the sum of the individual components’ activities [131]. By speeding up or slowing down the absorption of the primary therapeutic component, the combined activities of these drugs tend to boost its activity. Secondary metabolites derived from plants may improve the stability of active compounds or phytochemicals, reduce the occurrence of unwanted side effects, and have an additive, potentiating, or antagonistic impact [132]. It has been proposed that the enormous diversity of chemical structures found in these plants are specialized secondary metabolites involved in the organism’s relationship with the environment, such as pollinator attractants, signal products, defensive substances against predators and parasites, or pest and disease resistance [133].

Bitter substances that aid in the removal of waste products and pollutants, anti-inflammatory compounds that reduce swelling and pain, phenolic compounds that act as antioxidants and venotonics, antibacterial and antifungal tannins that act as natural antibiotics, and diuretic substances that aid in the removal of waste products and pollutants, as well as alkaloids that improve mood and provide a sense of well-being, are just a few examples [2,17,134]. Although some may believe that isolating phytochemicals and using them as single chemical entities is a better option, which has led to the replacement of plant extracts, a growing body of evidence suggests that using crude and/or standardized extracts rather than isolated single compounds may have some medical benefits [135].

## 9. Status, Causes and Challenges of Medicinal Plants

Concerns about natural resources, particularly plant species, continue to be a global problem [136]. Unfortunately, the rate of plant biodiversity loss is expected to accelerate [137]. According to available data, the severity of the situation is strongly pronounced in hot areas—the arid desert, as shown by the high number of plant species now under threat (critically endangered, endangered, or vulnerable). Multiple causes contribute to the loss of plant diversity, including habitat degradation, alien species competition, death from imported diseases, pollution, and overexploitation for a variety of purposes, including food, shelter, and medicine [136]. Due to the large number of species, the medicinal use of plants is one of the most important uses of natural resources [138].

Because of their substantial contributions to healthcare, financial income, cultural identity, and livelihood security, the value of medicinal plants is basically unlimited [139]. The Desert African medicinal plant business has recently been under increased attention as a result of rising concerns about the sustainability of wild plants, which are traditionally the primary source of traditional medicines [140]. The first stage in selecting species with conservation or resource management priorities is to gather information on the species that are sold, their prices, and the marketed volumes [141]. However, throughout Africa, a lack of scientific data on these critical components of the medicinal trade is prevalent and continues to be a serious issue [140]. Recent quantitative studies analyzing the quantities of medicinal plants traded have been reported for a few African nations, including South Africa [142], Ghana [141], Gabon [143], Tanzania [144], and Sierra Leone [145], as the relevance and utility of such data have grown. The information gained from this research will undoubtedly aid in the assessment of the conservation status of frequently used medicinal species.

Previous research has found that wild populations of medicinal plants are preferentially targeted, posing particular challenges for conservationists [146]. Their viability is determined by the kind of vegetation collected, relative abundance, and growth rates [147]. The vulnerability of medicinal plant species to overharvesting is also impacted by the life forms and plant parts utilized. Woody plants (shrubs and trees) make up the majority of the 51 most significant medicinal plants (65%). Sierra Leone [145], Gabon [143], South Africa [148], and Ghana [149] have all documented significant usage of non-sustainable organ harvesting. The removal of wood, bark, bulbs/corms, roots, or complete plants from any medicinal plant usually results in the death of an individual species [145]. On the other hand, harvesting leaves, fruits, or seeds, on the other hand, is typically seen as less harmful to plant survival.

Intensive pruning, on the other hand, may have an effect on the reproductive success of such plants. These concerns can be divided into four categories from an ecological standpoint: (1) growth, survival, and reproduction rates; (2) population structure and dynamics; (3) community structure and composition, plant-animal and plant-plant interactions; and (4) nutrient and organic matter dynamics, energy exchange [150]. As a result, while creating and assessing ways to minimize the negative impacts of medicinal plant overharvesting, researchers must consider the aforementioned processes (which correspond to distinct ecological levels).

## 10. The Cultivation of Medicinal Plants in Desert Areas a Solution?

The medicinal plant industry is extensive, comprising collectors/local sellers, and consumers, as well as heavily reliant sectors such as nutraceutical and pharmaceutical firms that require enormous volumes of raw material [140]. As a result, the existing conservation techniques have been impacted by the diversity of interested parties and the success of any future approaches/solutions. To alleviate the constant strain on medicinal plant populations in the wild, it is necessary to take rigorous steps to ensure the continuing availability of these precious natural resources [136,151,152]. Increased regulation and the adoption of sustainable wild gathering methods can provide enough protection for some species; however, a more feasible long-term solution may be an increase in desert medicinal plant cultivation [153].

Medicinal plant cultivation usually needs a lot of care and attention. Depending on the grade of medicinal plant components required, the circumstances and time of cultivation are different [154]. In comparison to Europe and Asia, medicinal plant production in Africa’s Saharan areas is still in its infancy. Strong attempts toward commercial medicinal plant production has recently been recorded in the northern (Egypt, Libya, Morocco, Tunisia), eastern (Uganda, Kenya, and Tanzania), western (Nigeria, Ghana and Sierra Leone), and southern Africa (South Africa and Madagascar) [153]. Cultivation initiatives for some of the most significant African medicinal plants are a priority among academics and policymakers. Furthermore, growing evidence from countries across the continents, such as South Africa [148] and Sierra Leone [145], strongly suggests that other equally valuable medicinal plants can be grown on a small and large scale with the right approach. Despite rare reports of medicinal plant growth, there is a distinct dearth of recorded information on the scale of such operations. As a result, additional research in this field is necessary.

**Table 1 molecules-28-01834-t001:** Medicinal plant used in Saharan regions, their habits, parts used, mode of administration, countries where they are used, toxicity, and ethnopharmacology.

Name of the Plant	Part(s) Used	Toxicity Studies	Mode of Administration	Traditional Use	Administration and Application Area	Country	Ref.
*RetamaRetam Web b.*	Fruits, seeds	Not toxic	Decoction, Infusion	Diabetes, Cold, skin wounds, back pain	Oral	Morocco, Algeria, Tunisia, Libya, Egypt, Palestina	[41]
*Genista saharae Cosson et Dur.*	Aerial part	Not toxic	Decoction	Infections of the respiratory system	Oral External use	Algeria, Libya, Morocco, Tunisia and Egypt	[52]
*Astragalus gombiformis Bomel.*	Different parts	Including toxic species	Decoction	Antidote against bites of snakes and scorpions	External use	Algeria, Libya, Morocco, Tunisia	[155]
*Eurphorbia guyoniana Bois et Reut.*	Aerial parts roots	Toxic	Decoction	Against venomous bites of scorpions antitussive, analgesic	Oral External use	Algeria, Tunisia, Libya and Morocco	[156]
*Ephedra alata DC.*	Aerial parts	Not toxic	Maceration	Cold, influenza, respiratory problems, hypertension	Oral External use	Asia, America, Europe, and North Africa	[157]
*Heliathemum lipii* (L.) *Pers.*	Aerial parts	Including toxic species	Maceration	Treat skin lesions common colds, as diuretic and for rheumatic	Oral External use	Asia and Africa	[158,159]
*Cyperus conglomeratus*	Aerial part	Not toxic	Maceration	Diuretic, analgesic and anthelmintic treatments	Oral External use	Africa, The Arabian Peninsula	[160]
*Calligonum comosum L’her.*	Aerial parts	Not toxic	Maceration Powder	Scorpion stings and snake bites	External use	Saudi Arabia, Algeria, Tunisia, Libya and Morocco	[161,162]
*Plantago albicans* L.	Seeds, leaves Powder	Not toxic	Infusion	Diabetes	Oral	Tunisia, Algeria and Libya	[41]
*Limoniastrum guyonianum Dur.*	Leaves	Not toxic	Decoction, infusion	Diabetes, scorpion stings and snake bites, anemia	Oral External use	Tunisia, Algeria and Libya	[41]
*Tamarix boveana*	Leaves	Not toxic	No reported	Diabetes, burn, illnesses of the kidney, diarrhea	Oral External use	Irano-Turanian, Mediterranean, Algeria	[163]
*Traganum nudatum Del.*	Aerial part	Not toxic	Decoction, powder	Diabetes, rheumatism, skin diseases, diarrhea	External use	Algeria	[41]
*Bassia muricata* (L.)	aerial parts	Not toxic	Powder	Analgesic, anti-inflammatory	External use	Iran, Palestine, North Africa.	[83]
*Atriplex halimus* L.	Leaves	Not toxic	Decoction	Diabetes, ovarian cysts, rheumatism, goiter	Oral	Morocco, Algeria, Tunisia, Libya	[164]
*Zygophyllum album* L.	Aerial part	Not toxic	Decoction	Diabetes	Oral	Morocco, Algeria, Tunisia	[165]
*Matricaria pubescens (desf) Schultz.*	Aerial part Leaves	Not toxic	Decoction, infusion	Diabetes	Oral	Morocco, Algeria, Tunisia	[41]
*Erodium glaucophyllum L’Her.*	Aerial part	Not toxic	Decoction	Oxytocic and astringent	Oral	Western Mediterranean coastal region	[166,167]
*Cleome arabica* L.	Aerial part	Toxic	Decoction, infusion	Diabestes, rheumatism	Oral	Distributed in the north of Africa	[168]
*Neurada procumbens* L.	Aerial par	Not toxic	Decoction, infusion	Antioxidant, diabetes, diarrhea	Oral	Sinai, Sudan, Ethiopia, Saudi Arabia	[91]

## 11. Conclusions and Recommendations

Based on extensive recent research in ethnopharmacology, green chemistry, and pharmaceutical chemistry, it demonstrates a renaissance in interest in plant medicines, including desert plants, for the treatment and prevention of many ailments. In hot, dry climates, medicinal plants continue to play an essential role in the rural healthcare system. Until now, the successful treatment of illnesses using plant products has not yet been thoroughly proven using rigorous scientific criteria to compete with the current conventional treatments; thus, a number of significant challenges must be overcome and addressed before their full potential can be realized.

The current review focused on the necessity for accurate recording of medicinal plants utilized by residents of the Saharan region to treat prevalent illnesses. The findings of this study revealed a rich diversity of medicinal plants used to treat various disease conditions as well as ethnomedicinal knowledge; as a result, if the trade of Saharan region herbal products is to increase, local laws must be TRIPS compliant, and issues of sustainable use and development of plant products must be addressed at the same time.

Studies have indicated that all medicinal plants that have never been reported previously yet have been utilized for millennia to cure a variety of severe ailments should be investigated for their unknown potential uses. Communities would surely benefit from further research and promotion of medicine as they work to preserve knowledge and incorporate particular techniques into healthcare services. This may open the door to further study by pharmacologists and phytochemists. Two things should be mentioned. First, local knowledge might be turned into pharmaceuticals or other products for sale, and the locals who are the guardians of this knowledge should be thanked and fairly compensated. Second, overexploitation of medicinal plants will inevitably jeopardize their existence, necessitating conservation efforts.

## Figures and Tables

**Figure 1 molecules-28-01834-f001:**
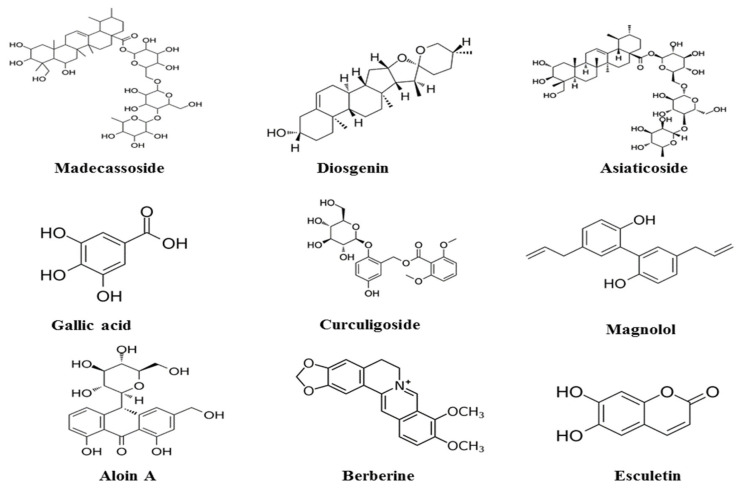
Chemical formulas for natural compounds.

**Figure 2 molecules-28-01834-f002:**
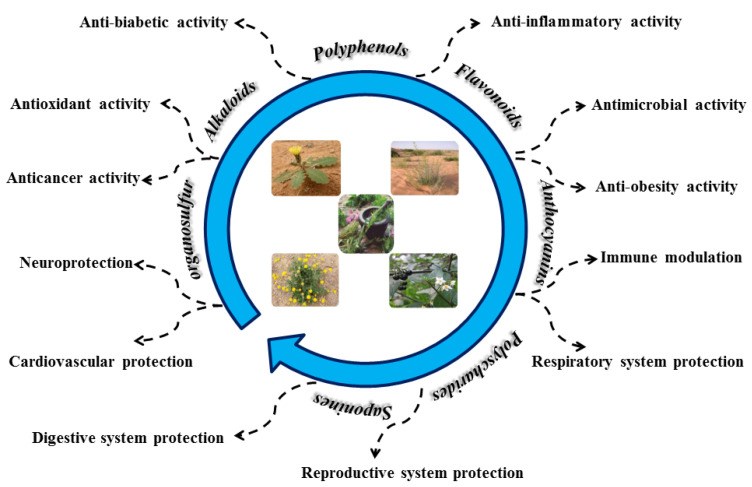
The most important biological activities of plant extracts.

**Figure 3 molecules-28-01834-f003:**
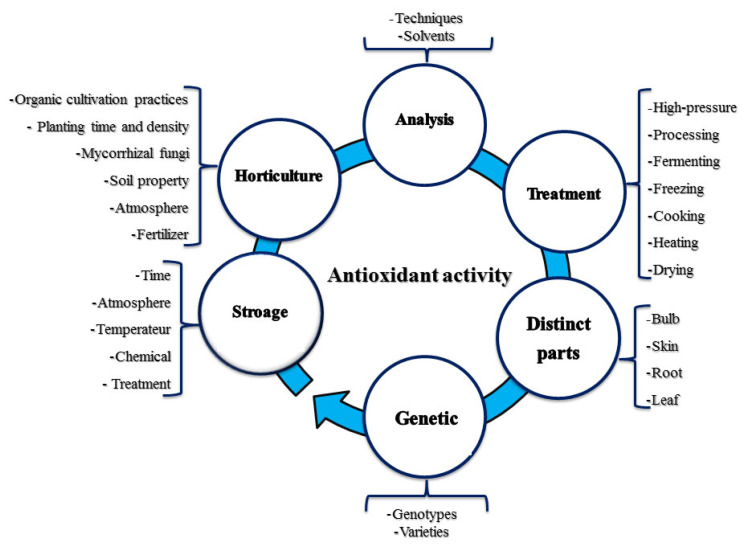
The most important factors affecting the effectiveness of antioxidants.

## Data Availability

Data are available in the manuscript.

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
