# Peer review of "Desert Endemic Plants in Algeria: A Review on Traditional Uses, Phytochemistry, Polyphenolic Compounds and Pharmacological Activities"

_molecules, 2023, doi:10.3390/molecules28041834_

Round 1

Reviewer 1 Report

Please capitalize the 'compound' in the title;

Please improve the format of figure 1.

Author Response

February 07th 2023

Journal: Molecule

Manuscript Number: Molecule-2130606

Desert Endemic Plants Under Extreme Climatic Conditions in Algeria : A Review on Traditional Uses, Phytochemistry, and Pharmacological Activities

Dear editor,

We gratefully appreciate the rigorous review of the referees who reviewed our manuscript, which added much strength and validity to our research. Some points really helped to improve the manuscript and we really appreciate that. Again, we thank all the reviewers for their valuable inputs, which have given the manuscript a chance to reach a satisfactory level for publication.

(The corrected parts can be identified with the tracking in the revised manuscript)

Our responses are as follows:

Reviewer #1

Comment 1

- Please capitalize the 'compound' in the title;

Response 1

We would like to thank the reviewer for this comment and thorough reading of this manuscript and for the thoughtful comments and constructive suggestions. Under the reviewer’s suggestion, We have corrected and improved our manuscript and we have corrected the title.

Comment 2

- Please improve the format of figure 1.

Response 2

We thank the reviewer for this comment, We are really sorry for this error and it was corrected.

We hope the Reviewer and the Editors will be satisfied with our responses to the ‘comments’ and the revisions for the original manuscript.

Thanks and Best Regards!

 Yours Sincerely,

Reviewer 2 Report

The work is interesting and up-to-date.

I have a number of remarks such as:

The abstract must be made according to the journal's template (Background, Methods..)

  At 5. Results and Discussion leave only Discussions.

In the case of a review, there are no results.

Please review!

Author Response

February 07th 2023

Journal: Molecule

Manuscript Number: Molecule-2130606

Desert Endemic Plants Under Extreme Climatic Conditions in Algeria : A Review on Traditional Uses, Phytochemistry, and Pharmacological Activities

Dear editor,

We gratefully appreciate the rigorous review of the referees who reviewed our manuscript, which added much strength and validity to our research. Some points really helped to improve the manuscript and we really appreciate that. Again, we thank all the reviewers for their valuable inputs, which have given the manuscript a chance to reach a satisfactory level for publication.

(The corrected parts can be identified with the tracking in the revised manuscript)

Our responses are as follows:

Reviewer#2

Comment 1

The work is interesting and up-to-date.I have a number of remarks such as:

Response 1

We would like to thank the reviewer for this comment and through reading of this manuscript and for the thoughtful comments and constructive suggestions.

Comment 2

- The abstract must be made according to the journal's template (Background, Methods..)

- At 5. Results and Discussion leave only Discussions.

- In the case of a review, there are no results.

- Please review!

Response2

We thank the reviewer for this comment and highlighting these points. We are really sorry for this error, and it was all corrected, as well as we have removed the word result.

We hope the Reviewer and the Editors will be satisfied with our responses to the ‘comments’ and the revisions for the original manuscript.

Thanks and Best Regards! 

Yours Sincerely,

Reviewer 3 Report

I think the title is too elaborate. The part "Under Extreme Climatic Conditions" in my opinion does not make sense because it is known what the climate is like in Algeria and on the other hand I did not find a reference in the manuscript regarding the impact of this climate on production or utilization efficiency in the context of the selected compounds used in the title, so I propose a new title: "Desert Endemic Plants in Algeria: A Review on Traditional Uses, Phytochemistry, Polyphenolic compounds and Pharmacological Activities".

Abstract: you use the abbreviation TRIPS-please use the full name first, only then use the abbreviation

Methodology: it seems to me that using unpublished dissertations and theses is not a good idea; using such material that has not gone through vetting, peer review does not make it a good source for support. In addition, please specify the timeframe for what period these searches were made, and shouldn't the keywords also include the name of the region for which this review is being done? Please take a peek at this publication for a methodical and bibliographic approach to such searches and review: https://doi.org/10.1007/s11356-020-08693-5.

Why specifically reviewed in the context of such compounds? After all, these plants certainly also accumulate heavy metals or PAHs, which are harmful and can inhibit their culinary or medicinal use. Please justify your choice of compounds for this review in the text of the manuscript.

Figure 1: Why collect and label the material if you didn't do research on them, but relied on publications as part of the fact that this is a review article? You just wanted to make sure that on these plants it was done according to the authors from the cited publications used for the review?

Results and Discussion: you write: "From the search above, more than 150 ethnobotanical articles, phytochem-istry and pharmacology papers were retained have been approved by this review." That is, it follows that you only used articles and Methodology states: "published and unpublished dissertations and the-ses, books, and articles", meaning also books. In that case, what final material was used to prepare this review. Please also provide the timeframe I previously inferred, from what period of years the material was collected for this review.

Author Response

February 07th 2023

Journal: Molecule

Manuscript Number: Molecule-2130606

Desert Endemic Plants Under Extreme Climatic Conditions in Algeria : A Review on Traditional Uses, Phytochemistry, and Pharmacological Activities

Dear editor,

We gratefully appreciate the rigorous review of the referees who reviewed our manuscript, which added much strength and validity to our research. Some points really helped to improve the manuscript and we really appreciate that. Again, we thank all the reviewers for their valuable inputs, which have given the manuscript a chance to reach a satisfactory level for publication.

(The corrected parts can be identified with the tracking in the revised manuscript)

Our responses are as follows:

Reviewer #3

Comment 1

I think the title is too elaborate. The part "Under Extreme Climatic Conditions" in my opinion does not make sense because it is known what the climate is like in Algeria and on the other hand I did not find a reference in the manuscript regarding the impact of this climate on production or utilization efficiency in the context of the selected compounds used in the title, so I propose a new title: "Desert Endemic Plants in Algeria: A Review on Traditional Uses, Phytochemistry, Polyphenolic compounds and Pharmacological Activities".

Response 2

We would like to thank the reviewer for this comment and through the reading of this manuscript and for the thoughtful comments and constructive suggestions. We have corrected these errors, also we have tried to find all the similar mistakes in the paper and have corrected them. So, as you suggested we have corrected the title of our manuscript as below:

"Desert Endemic Plants in Algeria: A Review on Traditional Uses, Phytochemistry, Polyphenolic compounds and Pharmacological Activities".

Comment 2

- Abstract: you use the abbreviation TRIPS-please use the full name first, only then use the abbreviation

Response 2

We thank the reviewer for this comment, We are really sorry for this error, and it was corrected.

So, we have add the mean of this Abbreviation in Abstract - (TRIPS):Trade Related Aspects of Intellectual Property Rights

Comment 3

- Methodology: it seems to me that using unpublished dissertations and theses is not a good idea; using such material that has not gone through vetting, peer review does not make it a good source for support. In addition, please specify the timeframe for what period these searches were made, and shouldn't the keywords also include the name of the region for which this review is being done? Please take a peek at this publication for a methodical and bibliographic approach to such searches and review: https://doi.org/10.1007/s11356-020-08693-5.

Response 3

We thank the reviewer for this comment and highlighting these points. We are sorry for these errors,

So, as suggested by the reviewer,

1-  The terms dissertations and unpublished dissertations have been deleted.

2- A time frame was set between 2000 and 2020.

3-Search keywords: medicine practitioner, traditional medicine, and traditional medicinal plant were used as sources of information.

4- we have added  this review to the list of references

Comment 4

- Why specifically reviewed in the context of such compounds? After all, these plants certainly also accumulate heavy metals or PAHs, which are harmful and can inhibit their culinary or medicinal use. Please justify your choice of compounds for this review in the text of the manuscript.

Response 4

We thank the reviewer for this comment,the selected plants in this paper are medicinal plants that grow spontaneously in the desert regions of Algeria, and it has been proven in traditional medicine that they are medicinal plants.

On the other hand in experimental studies about these plants, have been proven to be harmless (quantitative and qualitative analysis) they are harmless plants, even if they contain heavy metals or aromatic hydrocarbons, in small proportions, which is consistent with traditional medicine.

In future research on these plants, this paper will be an important and good reference article.

Comment 5

- Figure 1: Why collect and label the material if you didn't do research on them, but relied on publications as part of the fact that this is a review article? You just wanted to make sure that on these plants it was done according to the authors from the cited publications used for the review?

Response 5

We thank the reviewer for his/her comment, in the first figure these images of the plants studied are placed in order to facilitate their identification by the respected readers of the journal Molecules, in fact, for clarity.

Comment 6

- Results and Discussion: you write: "From the search above, more than 150 ethnobotanical articles, phytochem-istry and pharmacology papers were retained have been approved by this review." That is, it follows that you only used articles and Methodology states: "published and unpublished dissertations and the-ses, books, and articles", meaning also books. In that case, what final material was used to prepare this review. Please also provide the timeframe I previously inferred, from what period of years the material was collected for this review.

Response 6

We thank the reviewer for this comment, we have revised our manuscript, we have removed the terms dissertations and unpublished dissertations and we have also added the time frame in which these plants were studied which is between 2000 and 2020.

We hope the Reviewer and the Editors will be satisfied with our responses to the ‘comments’ and the revisions for the original manuscript.

Thanks and Best Regards!

Yours Sincerely,

Reviewer 4 Report

my advice is to re-write the article, to change the photos with new scientific professional ones, to reconsider the presented chemical formulas, classifiy them, in a word, put in order the informations that you present.

Author Response

February 07th 2023

Journal: Molecule

Manuscript Number: Molecule-2130606

Desert Endemic Plants Under Extreme Climatic Conditions in Algeria : A Review on Traditional Uses, Phytochemistry, and Pharmacological Activities

Dear editor,

We gratefully appreciate the rigorous review of the referees who reviewed our manuscript, which added much strength and validity to our research. Some points really helped to improve the manuscript and we really appreciate that. Again, we thank all the reviewers for their valuable inputs, which have given the manuscript a chance to reach a satisfactory level for publication.

(The corrected parts can be identified with the tracking in the revised manuscript)

Our responses are as follows:

Reviewer#4

Reviewer Comment

my advice is to re-write the article, to change the photos with new scientific professional ones, to reconsider the presented chemical formulas, classifiy them, in a word, put in order the informations that you present.

Authors response 

We would like to thank the reviewer for this comment and through reading of this manuscript and for the thoughtful comments and constructive suggestions. We are really sorry for this error and it was corrected. under the reviewer's suggestion, all typing and grammatical errors were corrected. Thank you for highlighting these points. As you requested, we have revised our manuscript as well as we have improved it. Dear reviewer, in fact, in recent years there are a few works on these plants, and for facility and give a good source about these plants studies, this review may be good for the students as well the research. which it will open new perspectives in the in future research works, especially in the field of biotechnology and separation.

We hope the Reviewer and the Editors will be satisfied with our responses to the ‘comments’ and the revisions for the original manuscript.

Thanks and Best Regards!

Yours Sincerely,

Reviewer 5 Report

This manuscript tried to provide a review on traditional uses, phytochemistry, polyphenolic compounds and pharmacological activities     for desert endemic plants under extreme climatic conditions in Algeria. However, the manuscript is badly written and structured, some main message gets lost thus many readers will get stuck in the hard to understand the content. There are some doubts on relationship between polyphenolic compounds and plants in Algeria. No Sufficient literature supported that polyphenolic compounds were main active phytochemical constituents of plants in Algeria. I strongly suggest that the authors rewrite the manuscript. References need to be rechecked and formatted according to Journal style.

Author Response

February 07th 2023

Journal: Molecule

Manuscript Number: Molecule-2130606

Desert Endemic Plants Under Extreme Climatic Conditions in Algeria : A Review on Traditional Uses, Phytochemistry, and Pharmacological Activities

Dear editor,

We gratefully appreciate the rigorous review of the referees who reviewed our manuscript, which added much strength and validity to our research. Some points really helped to improve the manuscript and we really appreciate that. Again, we thank all the reviewers for their valuable inputs, which have given the manuscript a chance to reach a satisfactory level for publication.

(The corrected parts can be identified with the tracking in the revised manuscript)

Our responses are as follows:

Reviewer#5

Comment 1

- This manuscript tried to provide a review on traditional uses, phytochemistry, polyphenolic compounds and pharmacological activities     for desert endemic plants under extreme climatic conditions in Algeria. However, the manuscript is badly written and structured, some main message gets lost thus many readers will get stuck in the hard to understand the content. There are some doubts on relationship between polyphenolic compounds and plants in Algeria. No Sufficient literature supported that polyphenolic compounds were main active phytochemical constituents of plants in Algeria. I strongly suggest that the authors rewrite the manuscript. References need to be rechecked and formatted according to Journal style.

Response 1

We would like to thank the reviewer for this comment and through reading of this manuscript and for the thoughtful comments and constructive suggestions. We are really sorry about some typo error and mistakes.

Under the reviewer's suggestion, all typing and grammatical errors were corrected.

In addition we have made a major revision of our paper.

We hope the Reviewer and the Editors will be satisfied with our responses to the ‘comments’ and the revisions for the original manuscript.

Thanks and Best Regards!

Yours Sincerely,

Round 2

Reviewer 3 Report

I would like to thank the Authors for all the answers. They are lucid, clear, transparent and exhaustive of the topic. As it stands, in my opinion, the work is already suitable for publication.

Reviewer 4 Report

no other suggestion for the authors

Reviewer 5 Report

The revised manuscript has been improved. However, references should be rechecked and formatted according to Journal style. The manuscript still needs a major revision.